# Cytotoxic Effect of Graphene Oxide Nanoribbons on *Escherichia coli*

**DOI:** 10.3390/nano11051339

**Published:** 2021-05-19

**Authors:** Shirong Qiang, Zhengbin Li, Li Zhang, Dongxia Luo, Rongyue Geng, Xueli Zeng, Jianjun Liang, Ping Li, Qiaohui Fan

**Affiliations:** 1Key Laboratory of Preclinical Study for New Drugs of Gansu Province, Institute of Physiology, School of Basic Medical Sciences, Lanzhou University, Lanzhou 730000, China; qiangshirong@lzu.edu.cn (S.Q.); lizhb16@lzu.edu.cn (Z.L.); zhangl_16@lzu.edu.cn (L.Z.); zengxl17@lzu.edu.cn (X.Z.); 2Northwest Institute of Eco-Environment and Resources, Chinese Academy of Sciences, Lanzhou 730000, China; gengrongyue18@mails.ucas.ac.cn (R.G.); liangjj@lzb.ac.cn (J.L.); liping@lzb.ac.cn (P.L.); fanqiaohui@nieer.ac.cn (Q.F.); 3College of Earth and Environmental Sciences, Lanzhou University, Lanzhou 730000, China

**Keywords:** graphene oxide nanoribbons, *E. coli*, cytotoxic effect, oxidative stress, cytomembrane damage

## Abstract

The biological and environmental toxicity of graphene and graphene derivatives have attracted great research interest due to their increasing applications. However, the cytotoxic mechanism is poorly understood. Here, we investigated the cytotoxic effect of graphene oxide nanoribbons (GORs) on *Escherichia coli* (*E. coli*) in an in vitro method. The fabricated GORs formed long ribbons, 200 nm wide. Based on the results of the MTT assay and plate-culture experiments, GORs significantly inhibited the growth and reproduction of *E. coli* in a concentration-dependent manner. We found that GORs stimulated *E. coli* to secrete reactive oxygen species, which then oxidized and damaged the bacterial cell membrane. Moreover, interaction between GORs and *E. coli* cytomembrane resulted in polysaccharide adsorption by GORs and the release of lactic dehydrogenase. Furthermore, GORs effectively depleted the metal ions as nutrients in the culture medium by adsorption. Notably, mechanical cutting by GORs was not obvious, which is quite different from the case of graphene oxide sheets to *E. coli*.

## 1. Introduction

Graphene is a material with a two-dimensional, monolayer, hexagonal carbon lattice structure. Due to its excellent physical, chemical, and mechanical properties, graphene and its derivatives have been extensively studied and have been used to prepare new composite materials [1,2,3,4]. Particularly, graphene oxide (GO), a graphene derivative, possesses both hydrophilic and hydrophobic oxygen-containing functional groups, such as hydroxyl, carboxyl, epoxy groups, thereby making graphene oxide soluble in water and a number of organic solvents [5]. Owing to its high solubility, GO has been widely applied in various biomedical applications, such as biological sensing, heat-based tumor targeting, drug delivery, and tissue engineering [4,6].

However, because GO is continuously discharged into the atmosphere, water, and soil, this has negative effects on the environment and human beings. Therefore, increasing attention has been paid to the toxic effects of GO and its derivatives. For instance, Audira reported that GO causes behavioral dysregulation in zebrafish [7]. Zhao et al. found that GO significantly induced oxidative stress and genotoxicity in *Eisenia fetida*, resulting in lipid peroxidation, decreased lysosomal membrane stability, and DNA damage [8]. Furthermore, in the bacterium *Azotobacter chroococcum*, GO induced cell death in a dose-dependent manner owing to membrane damage and oxidative stress [9]. Due to its high dispersibility and transformation and abundant functional groups, GO inhibited the growth of freshwater algae through indirect toxicity via the shading effect and nutrient depletion [10]. However, in green algae, cyanobacteria and diatoms, the toxicity of GO remains various and may be species-related [11]. Hence, the mechanism of GO toxicity might be diverse according to the species of model organisms.

Model organisms are important for the evaluation of GO toxicity. Suitable model organisms would not only be beneficial to the analysis of experimental results, but also could avoid the interference caused by biological factors. *Escherichia coli* (*E. coli*) is a Gram-negative bacterium that is commonly used as a model organism in nano-toxicology studies. As a prokaryote, *E. coli* has a simple structure. Therefore, GO can easily break its original ecological balance through interaction with *E. coli*, which is at the bottom of the biological chain [12].

The toxicity of GO and its derivatives using an *E. coli* model has been investigated. For instance, reduced GO exhibited a dose-dependent and time-dependent toxic effect on *E. coli* [13]. GO nanosheets showed an antibacterial activity to *E. coli* by generating reactive oxygen species (ROS) [14]. In addition, Baek et al. found that GO sheet significantly damages *E. coli* cell membranes due to its sharp edges [15]. Among graphene, graphite oxide, GO, and reduced GO, GO showed the highest antibacterial activity against *E. coli* [10], and this activity difference can be attributed to surface functionality [16]. The structure of a material determines its properties. Therefore, various fabrication methods for GO might result in different toxicities and mechanisms of interaction with *E. coli*.

Graphene oxide nanoribbons (GORs) are a derivative of GO and are mainly fabricated by longitudinally unzipping multi-walled carbon nanotubes (CNTs) [17]. Unlike the sheet structure of GO, GORs exhibit ribbon patterns, leading to more numerous zigzag edge sites occupied by various functional groups, such as hydroxy, carboxyl, and carbonyl groups. Due to its structure, GORs are characterized by excellent mechanical strength, easy solid–liquid separation, and high specific surface area and porosity [18], thus having various practical applications.

With increasing applications, more and more GORs has been intentionally and/or unintentionally released into the environment, which might have adverse effects on the microorganisms, animals and plants, and even human beings, in the ecological environment. However, there are only limited studies on the toxic effect of GORs on microorganisms, such as *E. coli*, and the underlying cytotoxicity mechanism is unclear. Hence, this study fabricated GORs via the Hummers method [19], investigated GORs toxicity to *E. coli*, and explored the underlying mechanism of its cytotoxic effect. Our results provide a theoretical basis for the toxicity mechanism of GO and its derivatives on microorganisms and the safe application of novel nanomaterials.

## 2. Materials and Methods

### 2.1. Strains and Reagents

The *E. coli* ATCC 25922 strain used in this study was obtained from the School of Nuclear Science and Technology, Lanzhou University (Lanzhou, China). GORs were prepared by longitudinally unzipping multi-wall carbon nanotubes as previously described [19,20]. The relative medium and chemicals employed were of analytical grade and were used directly.

### 2.2. Preservation and Culture of Strains

The Luria–Bertani (LB) liquid medium was prepared by dissolving 10 g tryptone, 10 g sodium chloride, and 5 g yeast extract in 1 L Milli-Q water, adjusted to pH 7.0, and sterilized at 120 °C for 20 min. To preserve the *E. coli* strain, 1 mL *E. coli* cultured overnight (OD_600_ = 0.1) was centrifuged at 1500 rpm for 5 min, and the supernatant was discarded. Then, 1 mL phosphate-buffer saline (PBS) containing 0.3 mL glycerin was immitted into sediment to seal *E. coli*; finally, sealed *E. coli* was stored at −80 °C for subsequent use.

### 2.3. Bacteriostatic Activity

The LB solid medium was prepared by adding 15 g agar powder to the LB liquid medium, adjusting pH to 7.5, and sterilizing it at 120 °C for 20 min. Then, it was evenly applied to the Petri dish before solidification.

*E. coli* cultured in the LB liquid medium overnight was diluted to OD_600_ = 0.1, and fabricated GORs (0, 20, or 200 µg/mL) were subsequently added to diluted *E. coil*. The mixture was cultured with shaking (140 rpm) for 2 h at 37 °C in a rotary shaker apparatus. After 1:100,000 dilution, 100 μL mixture was coated with the LB solid medium and cultured in a constant-temperature incubator at 37 °C for 24 h. Then, the number of colonies was counted and expressed as bacterial colony formation unit (CFU).

### 2.4. E. coli Survival

The 3-(4,5-Dimethyl-2-thiazolyl)-2,5-diphenyl-2-H-tetrazolium bromide (MTT) colorimetric assay was used to evaluate cytotoxicity. MTT is a yellow dye which can be reduced to the water-insoluble and blue-purple crystal formazan by succinic dehydrogenase produced by living cells [21]. Hence, the number of living cells is associated with the color intensity of formazan, which could dissolve in dimethyl sulfoxide (DMSO).

*E. coli* was incubated in the LB liquid medium at 37 °C overnight and subsequently treated with different concentrations of GORs (0, 20, 50, 100, and 200 µg/mL) for 24, 48, or 72 h in a constant-temperature incubator. *E. coli* was also treated with equal volumes of phosphate-buffer saline (PBS) at the same periods as control groups. After incubation, the samples were centrifuged at 10,000 rpm for 5 min and washed three times with PBS. Then, 200 µL MTT and 600 µL PBS were added to each sample, heated in a water bath at 37 °C for 30 min, and centrifuged at 10,000 rpm for 5 min. After the supernatant was discarded, 3 mL DMSO was added, and the mixture was centrifuged again at 10,000 rpm for 5 min. Finally, 200 μL supernatant was added to a 96-well plate, and absorbance was measured at 490 nm. Survival rate (IR; IR = experimental group/control group) was also calculated.

### 2.5. Oxidative Stress

Reactive oxygen species (ROS) produced by cells can react with unsaturated fatty acids to form lipid peroxidase, which can gradually decompose into a series of complex compounds, including malondialdehyde (MDA) [22]. Hence, the level of lipid oxidation can be indirectly determined by measuring MDA content. At high temperatures, MDA can react with thiobarbituric acid to form the reddish-brown trimethanol (3,5,5-trimethyl 2,4-dione), which has an absorption peak at 532 nm [23].

MDA content was determined to quantify oxidative stress. *E. coli* (OD_600_ = 0.1) exposed into different concentrations (0, 20, 50, 100, and 200 µg/mL) of GORs cultured with shaking (220 rpm) for 24 or 48 h at 37 °C was employed. Then, the mixture was centrifuged at 4000 rpm for 5 min and washed twice with PBS buffer. The supernatant was added to 1 mL MDA extract, subjected to ultrasound for 10 min (20% power), and centrifuged at 8000 rpm (4 °C) for 10 min. Then, the supernatant solution (1 mL) was mixed with 0.5 M hydrochloric acid (0.3 mL) and 1.0 mL TBA reagent in 0.67% and placed in a water bath at 95 °C for 30 min. After cooling to room temperature (25 ± 1 °C), the sample was added to 4.0 mL methanol–butanol solution, extracted for 45 min, and centrifuged for 10 min at 3000 rpm. Finally, 200 μL supernatant was added to a 96-well plate, and absorbance at 535 nm was measured. Normal saline was used as a blank control.

H_2_O_2_ content is also an indicator of cellular oxidative stress. Horseradish peroxidase (HRP) is a common peroxidase with high activity and stability [24]. Briefly, 90 mL of *E. coli* after 24 h culture was centrifuged at 8000 rpm for 10 min, washed three times with 30 mL PBS, 2 mL 0.9% NaCl was added, and homogenized manually for 10 min. Then, the samples were centrifuged at 8000 rpm (4 °C) for 10 min, and 200 µL liquid supernatant was extracted and reacted with 20 µL reaction solution (containing phenol red at 0.56 mmol/L and horseradish peroxidase at 17 µmol/mL) for 2 min. Afterwards, 10 µL 0.5 M NaOH was used to terminate the reaction, and absorbance was measured at 630 nm using a microplate reader. Milli-Q water was used as the negative control group.

### 2.6. Nutrient Depletion

Nanomaterials can adsorb nutrients in the medium. In GORs, this nutrient depletion can be determined by measuring the change in ion concentration in the medium. Briefly, GORs at various concentrations was added into the LB liquid medium, and the samples were cultured with shaking (220 rpm) at 37 °C for 96 h. Then, the mixtures were filtered twice using 0.22 nm filter paper. The supernatant was used to culture *E. coli*, and *E. coli* was counted after 24 h of cultivation. The concentrations of various ions, including Ca, Mg, K, Fe, Mn, Cu, and Zn, which are essential elements for *E. coli* growth, were determined using inductively coupled plasma mass spectrometry (ICP-MS). The LB liquid medium without GORs was used as the control.

### 2.7. Membrane Stability and Polysaccharides Uptake

The content of lactic dehydrogenase (LDH) could indirectly reflect the membrane stability. *E. coli* was exposed to GORs with serial culture system at different concentrations (0, 20, 50, 100, and 200 µg/mL) for 24 h. After washing twice with PBS buffer, LDH level was quantified using an LDH detection kit (Solarbio) according to the manufacturer’s protocol.

To investigate the affinity of GORs to the cellular surface of *E. coli*, we performed lipid-uptake experiments. The wavelength of polysaccharides at the maximum absorption peak was identified using a spectrophotometer. Polysaccharides at various concentrations (100, 200, 300, 400, 500, and 600 µg/mL) were prepared using the standard solution, and absorbance was measured. Then, different amounts of GORs were added, and the mixtures were shaken (180 rpm) at 37 °C for 24 h. Samples were centrifuged at 8000 rpm for 10 min, and the absorbance of the supernatant was measured.

### 2.8. Characterization

The morphologic changes in *E. coli* after interaction with GORs were observed by scanning electron microscopy (SEM) (Thermo, Waltham, MA, USA). Briefly, *E. coli* was treated with GORs at different concentrations (0, 50, and 100 µg/mL) for 24 h, washed twice with PBS buffer, fixed in 2.5% glutaraldehyde for 2 h at 4 °C, dehydrated with a series of ethanol (30%, 50%, 70%, 80%, 90% and 100%), and permeated with tert-butanol. Finally, these samples were freeze-dried, gold-plated, and subjected to SEM.

Transmission electron microscopy (TEM) (FEI, Hillsboro, OR, USA) was also employed, and sample preparation was similar to that in SEM, except that freeze-dried samples were dissolved in Mili-Q water by ultrasound and dripped to copper mesh.

Confocal laser scan microscopy (CLSM) (Zeiss, Oberkochen, Germany) was used to observe the changes in the membrane lipid bilayer of *E. coli*. *E. coli* samples exposed to GORs (0, 20, and 200 µg/mL) for 24 h were centrifuged at 10,000 rpm for 5 min, washed twice with PBS buffer, resuspended in 1 mL DiL dye in 10 μM/L preheated at 37 °C, and incubated at 37 °C for 20 min. After centrifugation at 1500 rpm for 5 min, the sediment was incubated with 1 mL LB liquid medium at 37 °C for 5 min twice and centrifuged at 1500 rpm for 5 min. The prepared samples were developed by CLSM at an excitation wavelength of 549 nm and an emission wavelength of 565 nm.

Fourier-transform infrared (FT-IR) spectroscopy (Bruker, Karlsruhe, Germany) was carried out using a Bruker Alpha instrument at 4000 to 400 cm^−1^. X-ray diffraction (XRD) (Rigaku, Tokyo, Japan) patterns were obtained on X′ Pert Pro Panalytical instrument equipped with a rotation anode using Cu–Kα radiation (40 kV and 30 mA). The scanning angle started from 3° (2θ) and continued to 80° with a step interval of 0.02° at a rate of 4.0°/min. Raman spectra were measured using a JY-HR 800 micro-Raman spectrometer (Jobin yvon, Longjumeau, Frence). All samples used for FT-IR, XRD, and Raman spectroscopy were prepared after incubation, centrifugation, and freeze-drying.

## 3. Results

### 3.1. Characterization of GORs and E. coli

In order to make sure GORs was successfully synthesized, TEM, Raman spectra, XRD and FT-IR were employed. The morphology of original GORs was observed via TEM (Figure 1a). It can be seen that the GORs were long ribbons with a width of 200 nm, which suggested linear edges and transparent nanosheets, and were different from graphene oxide nanotubes [25]. The Raman spectra of GORs (Figure 1b) showed characteristic D and G bands at ~1350 and 1570 cm^−1^, respectively. The XRD pattern (Figure 1c) has a sharp diffraction peak of [002] plane at 11° and a broad band at ~23°, indicating the disordered stacking of GORs. Based on the FT-IR spectra, GORs exhibited characteristic peaks at 3420 cm^−1^ (O–H stretching vibrations), 1730 cm^−1^ (C=O stretching vibrations), 1628 cm^−1^ (C=C stretching vibrations), 1232 cm^−1^ (C–OH stretching vibrations), 1050 cm^−1^ (C–O–C stretching vibrations), and 596 cm^−1^ (O–H bending vibrations) [20]. According to the SEM and TEM images (Figure 1e,f), *E. coli* cells were short, rod-like, and 1 µm in length and 0.5 µm in width.

### 3.2. Toxicity Effect of GORs on E. coli

After exposure to GORs in different concentrations, the mixture containing *E. coli* was coated with the LB solid medium flat and cultured for 24 h. The dose-dependent effect of GORs treatment on *E. coli* growth was clearly visible to the naked eye. Particularly, fewer *E. coli* plaque deposits were detected following exposure to higher concentrations of GORs (Figure 2a). The number of colonies decreased with the increasing GORs concentration (Figure 2b), suggesting that GORs are considerably toxic to *E. coli*. This is consistent with the findings of Jia et al. [26], who reported that carbon nanomaterials have dose-dependent noxious effect on alveolar macrophages. In addition, the inhibition effect of GORs on *E. coli* growth was similar to that of heavy metals and was dose-dependent, further indicating that GORs have a toxicity on *E. coli* [27].

The survival rate of *E. coli* also decreased with increasing GORs concentration (Figure 3), which is consistent with the results of bacteriostatic experiments. Overall, we demonstrated that GORs exert a dose-dependent toxic effect on *E. coli*.

### 3.3. Nutrient Depletion

GORs have a large specific surface area and many adsorption sites, which are used to remove contaminants from the environment [20]. Therefore, GORs exposed to the LB liquid medium might adsorb the nutrient element, and further influence the growth and reproduction of *E. coli*. In order to solve this issue, the content of nutrient elements (Fe, Zn, Ca, and Mg) in LB medium before and after being exposed to GORs was investigated.

As shown in Figure 4a, the contents of Fe and Zn in the supernatant significantly decreased upon exposure to 200 µg/mL GORs, and those of Ca and Mg also decreased after interaction with GORs, suggesting that GORs can adsorb Fe, Zn, Ca and Mg. Zn is widely found in a variety of proteins and involved in processes related to bacterial reproduction, such as DNA and RNA synthesis [28], DNA repair [29], virus-related protein generation [30], and antibiotic resistance generation [31]. Zn is also an important component of Cu–Zn superoxide dismutase, a metalloenzyme that plays important roles in eliminating excess reactive oxygen species [32]. Fe is an essential ingredient of RNA polymerase and RNA reductase, and is also necessary for the formation of cytochrome, hydrogenase, ferritin, and succinate dehydrogenase [33]. Thus, GORs removed the essential nutrients in the LB liquid medium via adsorption, thereby reducing their availability and inhibiting *E. coli* growth.

The supernatant of LB liquid medium after being exposed to GORs was further used to culture *E. coli* for confirming the influence of nutrient element removal by GORs on *E. coli* reproduction. As shown in Figure 4b, the survival rate of *E. coli* in the supernatant decreased with increasing concentrations of GORs. Thus, GORs inhibit the growth and reproduction of *E. coli* by competing for nutrients in the culture medium.

### 3.4. Oxidative Stress

Microorganisms generate ROS under stressful conditions [34]. Here, we used MDA and H_2_O_2_ as toxicological parameters to measure oxidative stress. As shown in Figure 5, MDA content in *E. coli* increased with exposure to increasing concentrations of GORs for 24 h and 48 h. This is consistent with the observed high MDA content upon the exposure of human RPMI 8226 cells to high concentrations of GO [35]. Moreover, high levels of MDA indicate high concentrations of ROS and increased membrane oxidation, suggesting that the cytotoxic effect of GORs is closely associated with increasing ROS levels.

Similarly, as shown in Figure 6, H_2_O_2_ content produced by *E. coli* increased after 24 h exposure to GORs at increasing concentrations. This result is consistent with that of the MDA experiment, suggesting that GORs cause cytotoxic effects on *E. coli* via ROS generation to oxidize the bacterial cell membrane.

### 3.5. Destruction of E. coli Membrane Components

LDH is a cytoplasmic enzyme and is released into the culture medium upon cell membrane damage. Thus, we analyzed LDH levels as an indicator of membrane damage (Figure 7). LDH concentrations slightly increased with GORs concentrations in a range of 0–50 µg/mL, suggesting that higher GORs concentrations induce more cell breakage, which led to more LDH release. However, LDH concentrations gradually decreased with GORs concentrations above 100 µg/mL, which might be due to the excessive dose of GORs inactivated LDH causing the low detection content of LDH in the system. However, the breakage of *E. coli* was confirmed via a TEM image of *E. coli* exposed to 200 µg/mL GORs, which was shown in Figure 8d.

The interaction of GO and its derivatives with microorganisms is complicated. Oxidative stress of E. coli caused by GORs can oxidize the lipid components of the membrane, thereby contributing to further cell damage. Baek et al. found that GO sheet significantly damages E. coli cell membranes due to its sharp edges [1,13], suggesting the mechanical cutting may also be a factor that causes E. coli damage. As seen in the TEM and SEM images, GORs contain sharp edges (Figure 1 and Figure 8a); however, mechanic cutting was not observed even at 200 µg/mL GORs concentration (Figure 8b,c), suggesting that mechanical cutting effect from GORs should be very limited to E. coli in this study. Tu et al. reported that graphene nanosheets can penetrate cell membranes and extract large amounts of phospholipids due to strong dispersion interaction between graphene and lipid molecules [36]. SEM (Figure 1e) and CLSM images of E. coli (Figure 9a) showed that E. coli was evenly distributed in the control groups. However, upon treatment with GORs, E. coli aggregated around the GORs (Figure 9b–e). This was possibly due to the special affinity of GORs to E. coli membrane, resulting in the accumulation of E. coli cells around GORs. This affinity may lead to the adsorption of the membrane components by GORs, thereby increasing their bacteriostatic potency.

Polysaccharides are an abundant and important membrane component of *E. coli*. Thus, we investigated their interaction with GORs (Figure 10). We found that the solid phase concentration of polysaccharides at 200 µg/mL GORs was always higher than that at 20 µg/mL GORs, indicating that GORs adsorbed the polysaccharide components of the *E. coli* membrane. Therefore, the adsorption of polysaccharides may be another important mechanism for *E. coli* membrane damage, which caused LDH leakage.

## 4. Conclusions

Cytotoxic effect of GORs on *E. coli* and its underlying mechanism was explored in an in vitro method. GORs were long ribbons in 200 nm wide characterized using TEM. Plate-culture experiments showed that GORs exerted a significant cytotoxic effect on *E. coli* in a concentration-dependent manner. The depletion of nutrients in the LB medium showed that GORs adsorbed metal ions and competed with *E. coli*, which inhibit the growth and reproduction of *E. coli*. Based on increased MDA and H_2_O_2_ levels in the culture media, GORs stimulated *E. coli* to secrete ROS, which can oxidize and damage the bacterial membrane. Moreover, SEM and CLSM images revealed that *E. coli* aggregated around GORs, resulting in the adsorption of membrane polysaccharides by GORs to further cause membrane damage and LDH release. Notably, mechanical cutting was not obvious, which is quite different from the study between GO sheets and *E. coli*.

## Figures and Tables

**Figure 1 nanomaterials-11-01339-f001:**
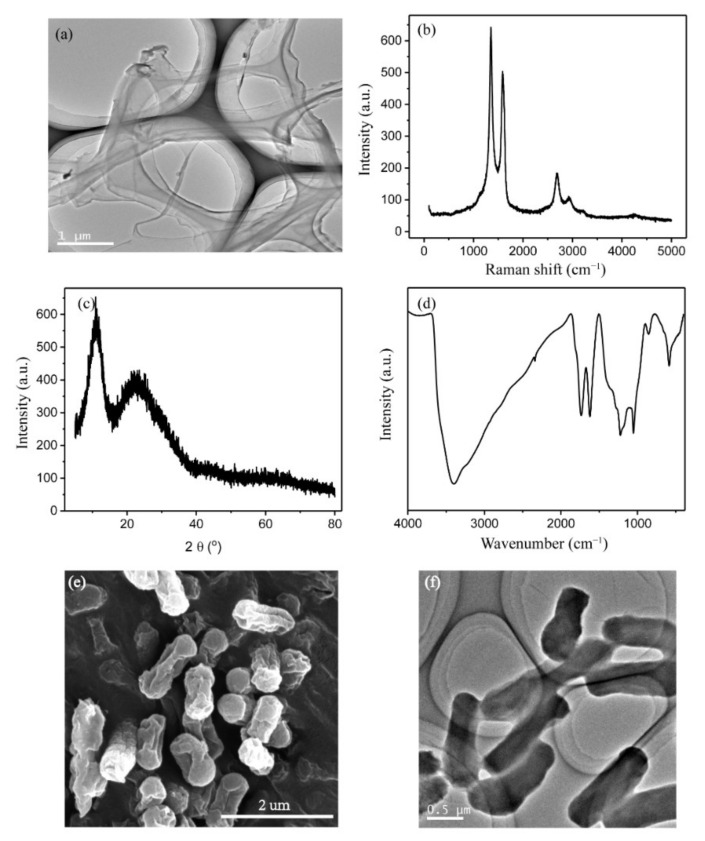
The TEM (**a**), Raman spectra (**b**), XRD spectra (**c**), and FT-IR spectra (**d**) of GORs, and SEM (**e**) and TEM (**f**) of *E. coli*.

**Figure 2 nanomaterials-11-01339-f002:**
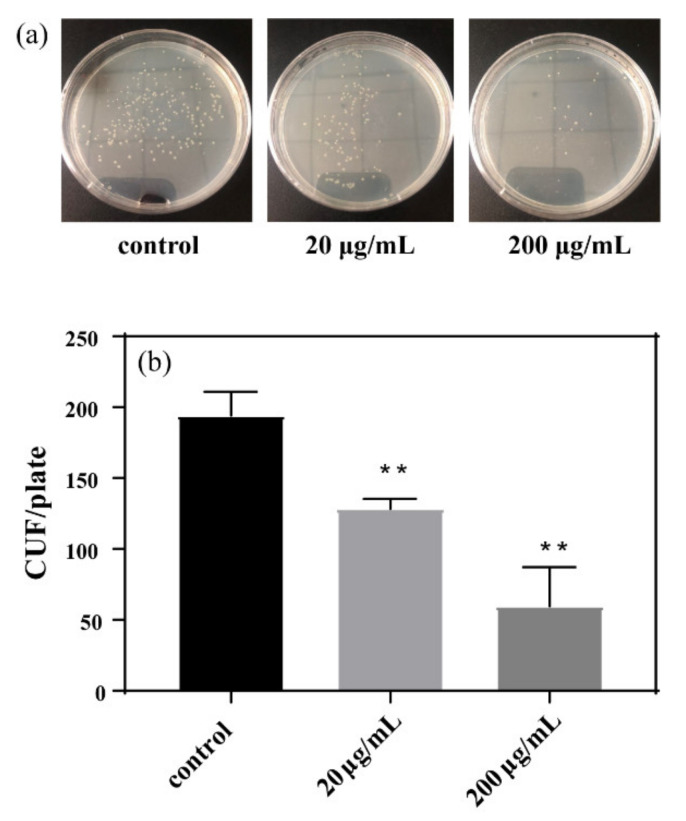
The result of bacteriostatic experiments after 24 h of culture (**a**) and the count of plaque of *E. coli* (**b**) (The initial *E. coli* solution OD_600_ is 0.1) (** *p* < 0.01).

**Figure 3 nanomaterials-11-01339-f003:**
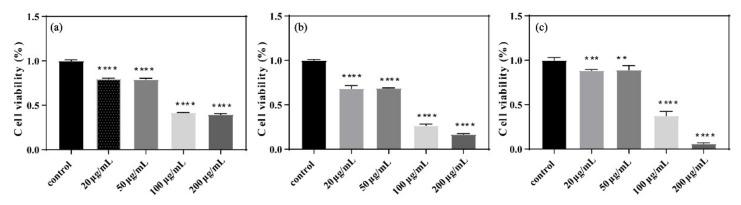
Survival rates of *E. coli* exposed to GORs in different concentrations at 24 h (**a**), 48 h (**b**), and 72 h (**c**) (** *p* < 0.01, *** *p* < 0.001 and **** *p* < 0.0001).

**Figure 4 nanomaterials-11-01339-f004:**
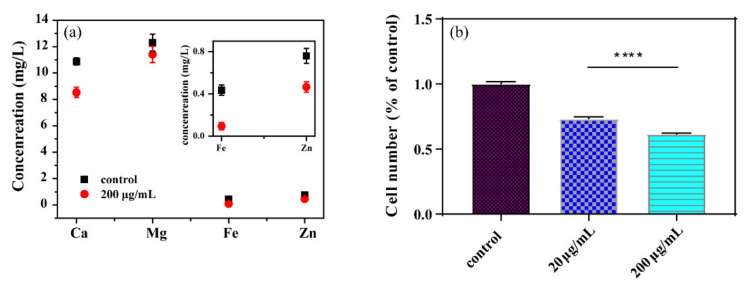
The content of Ca, Mg, Fe and Zn in supernatant of LB liquid medium after interaction with 0 and 200 µg/mL GORs (**a**), and the *E. coli* number after culture in supernatant after interaction with 0, 20 and 200 µg/mL GORs (**b**) for 24 h (**** *p* < 0.0001).

**Figure 5 nanomaterials-11-01339-f005:**
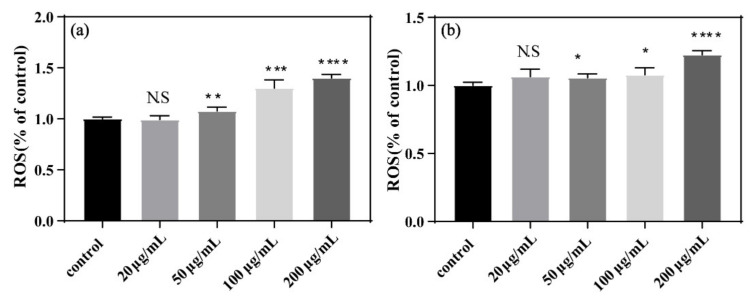
MDA content of *E. coli* exposed to GORs at different concentrations of GORs for 24 h (**a**) and 48 h (**b**) (* *p* < 0.05, ** *p* < 0.01, *** *p* < 0.001 and **** *p* < 0.0001).

**Figure 6 nanomaterials-11-01339-f006:**
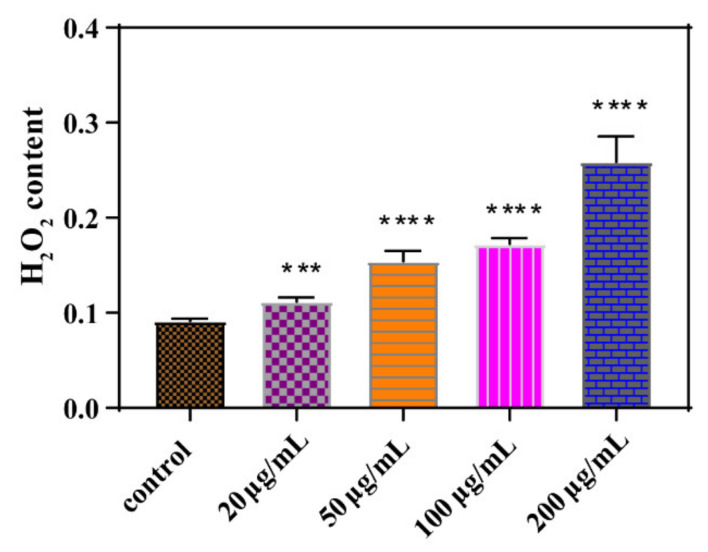
Peroxidase content of *E. coli* exposed to GORs in different concentrations (*** *p* < 0.001, **** *p* < 0.0001).

**Figure 7 nanomaterials-11-01339-f007:**
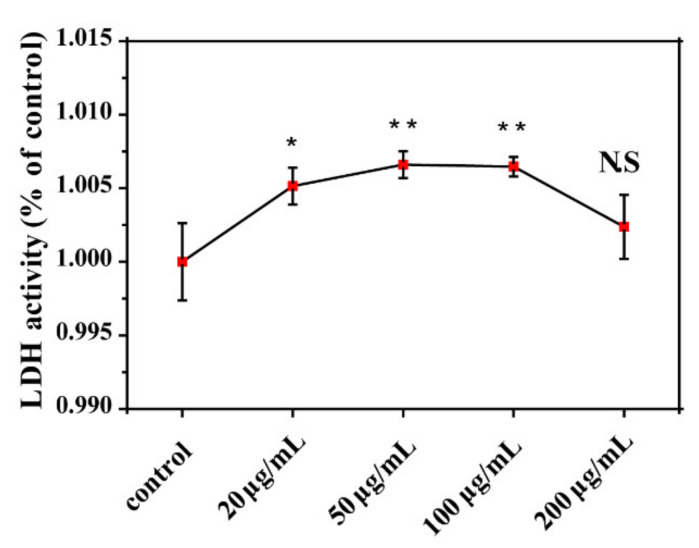
LDH content of *E. coli* exposed to GORs in different concentrations for 24 h (* *p* < 0.05, ** *p* < 0.01).

**Figure 8 nanomaterials-11-01339-f008:**
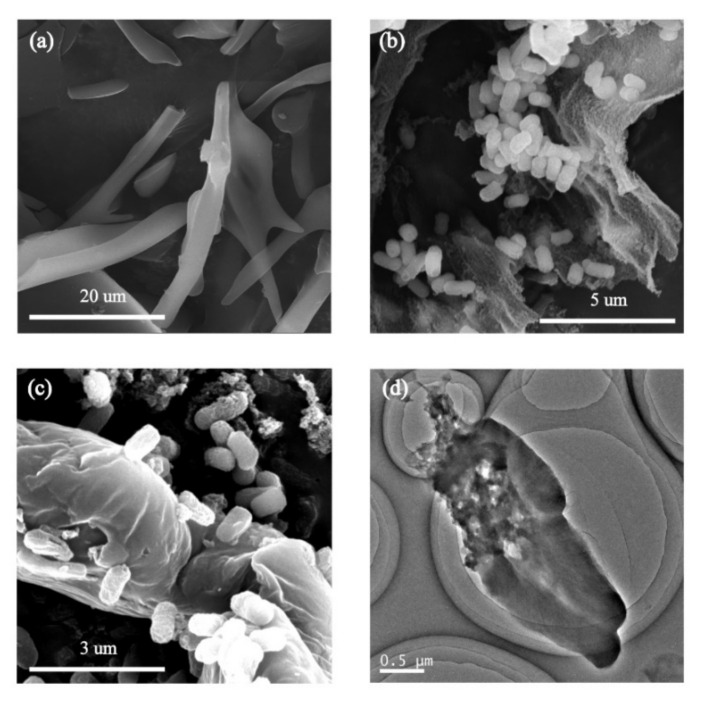
SEM of GORs (**a**) and *E. coli* exposed to GORs in 50 µg/mL (**b**) and 200 µg/mL (**c**), and the TEM of *E. coli* exposed to GORs in 200 µg/mL (**d**).

**Figure 9 nanomaterials-11-01339-f009:**
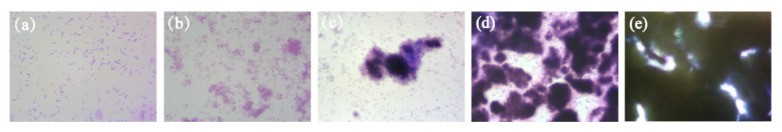
CLSM of *E. coli* exposed to GORs in 0 µg/mL (**a**), 20 µg/mL (**b**), 50 µg/mL (**c**), 100 µg/mL (**d**) and 200 µg/mL (**e**).

**Figure 10 nanomaterials-11-01339-f010:**
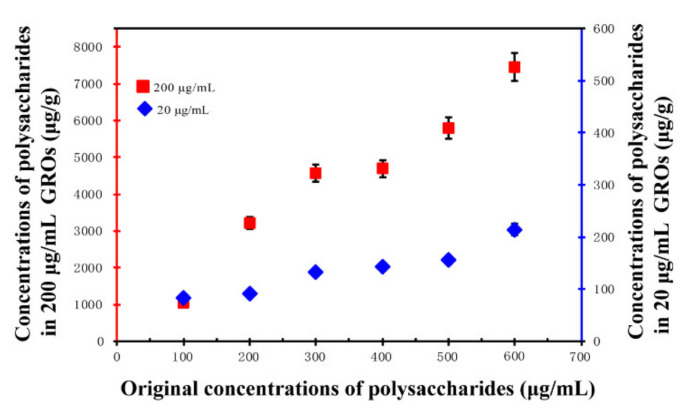
Changes in solid phase concentration of polysaccharides in 20 µg/mL and 200 µg/mL GORs (error bars stand for standard deviation).

## Data Availability

The data presented in this study are available on request from the corresponding author.

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
