# Peer review of "Cytotoxic Effect of Graphene Oxide Nanoribbons on Escherichia coli"

_nanomaterials, 2021, doi:10.3390/nano11051339_

Round 1
Reviewer 1 Report
In the manuscript entitled “Cytotoxic effect of graphene oxide nanoribbons on Escherichia coli” the authors investigated the cytotoxic effect of graphene oxide nanoribbons (GORs) on Escherichia coli (E. coli) through in vitro methods. Based on the results of the experiments, they found that: GORs significantly inhibited the growth and reproduction of E. coli in a concentration-dependent manner; GORs stimulated E. coli to secrete reactive oxygen species, which can oxidize and damage the bacterial cell membrane; GOR effectively depleted the metal ions as nutrients in the culture medium by adsorption; interaction between GORs and E. coli cytomembrane resulted in polysaccharide adsorption by GORs and the release of lactic dehydrogenase.
The topic of the present study is important and worth of more study. More discussions on the results are needed in order to explain better the cytotoxicity mechanism of GORs.
There are some issues to be changed:
- the importance of the work should be addressed more clear in the introduction section.
-line 16: “Here, we investigated the cytotoxic effect of graphene oxide nanoribbons (GORs) on Escherichia coli (E. coli) by in vitro methods” instead of “Here, we investigated the cytotoxic effect of graphene oxide nanoribbons (GORs) on Escherichia coli (E. coli) in vitro method.”
-line 51: “suitable model” instead of “suitable modal;
- line 75: a reference is required after “Hummers method”;
- In Materials and Methods section, the authors should provide at least a brief description of the method of synthesis of GORs. They stated at line 82: “GORs were prepared by longitudinally unzipping multi-wall carbon nanotubes as previously described [17]”, but reference [17] does not describe this method.
- line 206: “E. coli cells were short rod-like of 1 μm length and 0.5 μm width” instead of “E. coli cells was short 206 rod-like with a length in 1 μm and a width in 0.5 μm”.
- line 212: “the mixture containing E. coli was coated with the LB solid medium….” instead of “the mixture containing E. coli was 212 coated to the LB solid medium”
- line 234: “nutrient elements” instead of “nutrient element”
- line 278: The authors stated: “Thus, we analyzed LDH levels as an indicator of membrane damage (Fig. 7). LDH concentrations gradually increased with GORs concentrations, suggesting that higher GORs concentrations induce more cell breakage, which leaded to more LDH release.” This statement is not sustained by the data presented in Fig. 7. I recommend the authors to re-check the data!
-Fig. 1e is the same as Fig. 8a. One of them should be removed.
-Fig. 1f is the same as Fig. 8e. One of them should be removed.
-Line 321: “inhibits” instead of “inhabits”.
Reviewer 2 Report
In this manuscript, the author reports ‘Cytotoxic effect of graphene oxide nanoribbons on Escherichia coli’. The manuscript is well discussed. There are some minor comments which is required before getting a possible publication
- The scale bars in Fig.8c, Fig.8d and legends of Fig.10 are not clearly visible.
- The author should write purpose for each test in one/two sentences (in brief) before explaining the results of the characterization techniques. Therefore, the logic and organization of this part will be enhanced.
- The formatting and grammatical errors in the article need to be checked carefully.
- What stands for error bars in Fig.10? It should be mentioned in the Figure caption.
- Conclusion should be compact and precise.
- The authors cited some of the relevant research works that have been conducted in this area however there are few that need to be included (shown below) in the Introduction section: Ultrasonics Sonochemistry39 (2017): 577-588, Fibers and Polymers6 (2019): 1161-1171
